# Surgical capacity, productivity and efficiency at the district level in Sub-Saharan Africa: A three-country study

Mengyang Zhang[1]*, Jakub Gajewski[1], Chiara Pittalis[1], Mark Shrime[1], Henk Broekhuizen[2], Martilord Ifeanyichi[2], Morgane Clarke[1], Eric Borgstein[3], Chris Lavy[4], Grace Drury[4], Adinan Juma[5], Nyengo Mkandawire[3], Gerald Mwapasa[3], John Kachimba[6], Michael Mbambiko[7], Kondo Chilonga[8], Leon Bijlmakers[2], Ruairi Brugha[9]

1 Institute of Global Surgery, Royal College of Surgeons in Ireland, Dublin, Ireland, 2 Department for Health Evidence, Radboud University Medical Centre, Nijmegen, The Netherlands, 3 Department of Surgery, University of Malawi College of Medicine, Blantyre, Malawi, 4 Nuffield Department of Orthopaedics, Rheumatology and Musculoskeletal Sciences, University of Oxford, Oxford, United Kingdom, 5 East Central and Southern Africa Health Community, Arusha, United Republic of Tanzania, 6 Department of Surgery, Surgical Society of Zambia, University of Zambia University Teaching Hospital, Lusaka, Zambia, 7 Zambia College of Medicine and Surgery, Lusaka, Zambia, 8 Department of Surgery, Kilimanjaro Christian Medical Centre, Moshi, Tanzania, 9 Department of Epidemiology and Public Health Medicine, Royal College of Surgeons in Ireland, Dublin, Ireland

* mengyangzhang@rcsi.com

**Data Availability Statement:** All relevant data are within the paper and its Supporting Information files.

## Abstract

### Introduction

Efficient utilisation of surgical resources is essential when providing surgical care in low-resources settings. Countries are developing plans to scale up surgery, though insufficiently based on empirical evidence. This paper investigates the determinants of hospital efficiency in district hospitals in three African countries.

### Methods

Three-month data, comprising surgical capacity indicators and volumes of major surgical procedures collected from 61 district-level hospitals in Malawi, Tanzania, and Zambia, were analysed. Data envelopment analysis was used to calculate average hospital efficiency scores (max. = 1) for each country. Quantile regression analysis was selected to estimate the relationship between surgical volume and production factors. Two-stage bootstrap regression analysis was used to estimate the determinants of hospital efficiency.

### Results

Average hospital efficiency scores were 0.77 in Tanzania, 0.70 in Malawi and 0.41 in Zambia. Hospitals with high efficiency scores had significantly more surgical staff compared with low efficiency hospitals (DEA score<1). Hospitals that scored high on the most commonly utilised surgical capacity index were not the ones with high surgical volumes or high efficiency. The number of surgical team members, which was lowest in Zambia, was strongly, positively correlated with surgical productivity and efficiency.

**Funding:** The project is funded by the European Union's Horizon 2020 research and innovation programme under the grant agreement No. 733391. Salaries of JG, MZ, CP, HB, MC, AG, and GM were paid by this grant. Cost of the study was covered by this grant. The funders had no role in study design, data collection and analysis, decision to publish, or preparation of the manuscript.

**Competing interests:** The authors have declared that no competing interests exist.

## Conclusion

Hospital efficiency, combining capacity measures and surgical outputs, is a better indicator of surgical performance than capacity measures, which could be misleading if used alone for surgical planning. Investment in the surgical workforce, in particular, is critical to improving district hospital surgical productivity and efficiency.

## Introduction

National Surgical Obstetric and Anaesthesia Plans (NSOAP) have been launched in Zambia [1] and Tanzania [2] to provide comprehensive roadmaps towards improving national surgical systems. However, the empirical basis for these plans, especially with respect to the capacity needs of surgical systems is limited [3], undermining the potential for translating the plans into feasible and sustainable national surgical strategies [3]. With the increasing emphasis on widening access to surgical services, there have been efforts to support and improve surgical capacity [4–7], and to develop tools and indicators to monitor progress [8]. These have addressed adequacy of human resources, supplies and infrastructure, the main components of (surgical) capacity, with a view to improving surgical productivity [5, 6, 9]. However, what has been lacking is a systematic study of the surgical system factors that determine surgical efficiency, productivity, and the relationship between the two at district hospitals in Sub-Saharan Africa (SSA). Hospital efficiency and productivity measures may have greater potential for informing scalable strategies for achieving sustainable improvements in surgical output.

There are different published methods and tools for assessing surgical capacity [8, 10]. These include the World Health Organisation (WHO) Tool for Situational Analysis to Assess Emergency and Essential Surgical Care [11] and the WHO Service Availability and Readiness Assessment Tool [12]. Both include long surveys and lack a clear and easy way to standardise indices for reporting survey results [13]. Qualitative methods, which are more time-consuming (or labour intensive) have a similar limitation in that surgical capacity is hard to capture in a uniform manner across different settings [10]. The Personnel, Infrastructure, Procedures and Supplies (PIPES) survey, which captures resource availability in different domains, is a capacity assessment tool widely used in low- and middle-income countries (LMICs) [14–16]. It contains fewer questions than the WHO tools, and provides a simple quantitative assessment tool for overall capacity and resource availability [17].

On the other hand, a growing number of publications, including health sector studies, have used the data envelopment analysis (DEA) method to undertake efficiency analysis [18]. Advantages of this non-parametric method include informative efficiency measurements, which can control multiple input and output variables in the analysis; reliable identification of inefficiency; and generation of evidence about productivity and efficiency of resource use [19–21]. DEA evaluates the relative efficiency of a decision-making unit (a hospital in this study) in relation to its peers and is not affected by multicollinearity between variables.

District level hospitals (DLHs) are the frontline providers of accessible, essential surgical care for rural populations in SSA [22]. However, the factors that drive surgical productivity and efficiency at the district level are largely unknown. This paper aims to explore the factors associated with hospital efficiency and estimate the relationship between surgical capacity, productivity, and efficiency of district hospitals in three SSA countries. The analysis is based on district hospital data, collected through the Scaling up Safe Surgery for District and Rural Populations in Africa (SURG-Africa) project [23].

# Materials and methods

## Ethics approval

Prior Ministry of Health approval for data collection and informed audio-recorded consent for interviews from respondents were obtained. All of the approving research ethics committees (RECs) waived the requirement for writ-ten informed consent. Ethical approval was granted by the REC of the Royal College of Surgeons in Ireland, the project consortium lead, under approval No. REC 1417. In the implementation countries, ethical approval was received from the College of Medicine Research Ethics Committee in Malawi (approval No. P.05/17/2179), the University of Zambia Biomedical Research Ethics Committee (approval No. 005-05-17), the Kilimanjaro Christian Medical College Research Ethics and Review Committee (approval No. CRERC 2026), and the National Institute for Medical Research in Tanzania (approval No. NIMR/HQ/R.8a/Vol. IX/2600).

## Setting

This retrospective cohort study involved eighteen district hospitals in Malawi, twenty-two in Tanzania, and twenty-one in Zambia. The data on the surgical capacity of the 61 district hospitals were collected in July-November 2017 as part of a situation analysis for SURG-Africa, using the PIPES survey and a custom-made survey to supplement data missing in the original PIPES [23]. Details of three months of surgical operations, including types and numbers of main procedures, were collected from district hospital theatre logbooks between January and March 2018.

## Measurement

**Surgical capacity.** The PIPES Surgical Capacity Assessment tool measures the availability of five production factors: personnel, infrastructure, procedures, equipment and supplies (Table 1). The personnel section of the original PIPES tool captured only the availability of surgical specialists, anaesthesiologists, medical doctors doing surgery and nurse anaesthetists [24]. After consultations with designers of the tool this was extended to include all active surgical and anaesthesiology providers, notably non-physician clinicians (NPCs) capable of performing caesarean sections (CSs) and non-physician anaesthesia providers (NPAPs). This allows a more accurate picture of surgical care providers at the district level, since NPCs play a key role in surgical service delivery in the three countries under study [25]. Thereby, the number of variables in the PIPES tool increased from 105 to 107. The procedures section of the PIPES tool captures the types of surgery that the hospital team is able to undertake, rather than the numbers of procedures undertaken in the DLHs, which were captured separately. The other three inputs quantify the availability of essential infrastructure, equipment and supplies.

**Table 1. Composition of the PIPES capacity index.**

| Production factors | Content | Max. score |
|---|---|---|
| **Personnel** | The total number of general surgeons, anaesthesiologists, medical doctors, nurse anaesthetists, NPCs doing surgery and NPAPs administering anaesthesia | No limit |
| **Infrastructure** | The number of items always available and functioning operating rooms | No limit |
| **Procedure** | The self-declared range of procedures (from the list) that could be conducted | 40 |
| **Equipment** | The number of items always available | 22 |
| **Supplies** | The number of items always available/sufficient | 25 |

The PIPES overall score is calculated using the following formula:

$$PIPES\ index = \frac{sum\ of\ items\ available}{107} \times 10$$

**Surgical productivity.** For this paper, surgical productivity of a district hospital was defined as the number of operations performed within a given timeframe. To ensure comparability across hospitals and countries, the numbers of six surgical procedures were counted: hernia repair, hydrocele repair, hysterectomy, laparotomy, CSs, and surgical management of fracture in the operating room, which were the most frequently conducted major operations at the district level [26].

**Hospital efficiency.** Hospital efficiency was defined as the ability of a hospital to conduct the maximum possible number of operations given the availability of each of the five production factors. The efficiency score was calculated using DEA (see S1 File). The aim was to compare the surgical volume that each hospital actually conducted with its theoretical maximum volume–the production/possibility frontier. Hence an output-oriented model with the assumption of constant returns to scale was selected, based on the model proposed by Charnes et al. [27]. Efficiency results using other model settings [28], including variable returns to scale and nonincreasing returns to scale [29], were also calculated. Details can be found in S1 Table.

Hospital efficiency can be expressed as the ratio between the weighted sum of outputs and the weighted sum of inputs. The value of the efficiency score ranges from 0 (total inefficiency) to 1 (maximum efficiency). The output variable in the DEA was the combined volume of the six surgical procedures of interest. Input variables were the available personnel, infrastructure, procedures, equipment and supplies, i.e. the PIPES production factors. The common-set of weights (CSW) method was used to calculate the optimal input weights. A more detailed outline of the calculation of the production factors' weights used in the PIPES tool and calculated by the DEA is presented in S1 File.

## Analysis

To compare different indicators of hospital performance, a graph was created to demonstrate the ranking of a district hospital vis-a-vis its country peers. Hospitals were ranked from highest to lowest for each of the three indicators: the PIPES capacity index, numbers of surgical procedures, and efficiency scores; with colours in Fig 3 representing high (green), medium (yellow) and low (red) values for each indicator. The relationships between the indicator ranks are shown graphically by linked lines.

The relationship between surgical productivity and PIPES inputs was estimated using the quantile regression model at the 5th, 50th and 75th percentiles. Two tests with and without country dummy variables were included to ensure the results are consistent after controlling for country specific characteristics. Tanzania was set (arbitrarily) as the reference country. Standard errors were clustered at the hospital and country levels.

Two follow-up tests to strengthen model specification [19], estimating the relationship between hospital efficiency and production factors using a Tobit regression and two-stage bootstrap regression [30], are detailed in S4 Table.

The statistical analysis was conducted using Stata 16SE [31]. R package rDEA was used to generate the DEA score and the weights of input variables. The CSW-DEA was calculated using self-writing R code. Descriptive graphs were created using Stata 16SE and Tableau Desktop 2020.

**Table 2. Mean comparison of surgical capacity and production factors by country.**

|  | All | | Tanzania | | Malawi | | Zambia | |
|---|---|---|---|---|---|---|---|---|
|  | Mean | Std. Dev. | Mean | Std. Dev. | Mean | Std. Dev. | Mean | Std. Dev. |
| **PIPES** | 7.6 | 1.0 | 7.2 | 1.0 | 8.2 | 0.8 | 7.2 | 0.9 |
| **Personnel** | 11.4 | 7.8 | 10.3 | 5.3 | 18.7 | 6.6 | 4.7 | 2.3 |
| **Doctors*** | 3.2 | 3.0 | 6.2 | 3.3 | 1.0 | 1.0 | 3.0 | 1.8 |
| **NPC** | 8.4 | 8.9 | 3.7 | 2.3 | 18.1 | 7.8 | 2.2 | 1.1 |
| **Infrastructure** | 8.6 | 1.7 | 9.1 | 1.1 | 7.5 | 2.0 | 9.3 | 1.3 |
| **Procedures** | 26.7 | 4.2 | 25.3 | 4.9 | 28.7 | 2.9 | 25.7 | 3.9 |
| **Equipment** | 17.5 | 3.2 | 16.2 | 3.7 | 17.2 | 2.8 | 18.8 | 2.6 |
| **Supplies** | 16.8 | 5.3 | 16.7 | 6.5 | 15.4 | 3.7 | 18.5 | 5.3 |
| **N** | 61 | | 18 | | 22 | | 21 | |

Note

* refers to the number of general surgeons, anaesthesiologists, medical doctors, and nurse anaesthetists.

## Results

### Surgical capacity (PIPES index)

The average value of the PIPES overall surgical capacity scores by country and subscores for each of the production factors are presented in Table 2. Malawi has the highest overall capacity score, mainly because of the higher number of surgical personnel, the majority of whom are NPCs, compared to the other two countries. The average subscores for infrastructure and supplies are lowest in Malawi. Hospitals in Zambia have better infrastructure, equipment and supplies than in the other two countries, but considerably fewer surgical personnel. Hospitals in Tanzania have the lowest subscores for surgical procedures and equipment availability.

**Surgical productivity.** The volumes of the six index surgical procedures performed, by country, are reported in Table 3. CS is the most commonly conducted procedure at district level hospitals in all three countries, ranging from 71% (980/1377) of the index procedures in Zambia to 85% (1902/2235) in Tanzania. These are followed by hernia repairs, which in Malawi account for twice the average volume of procedures recorded in the other countries.

**Table 3. Mean comparison of surgical productivity and most commonly surgical procedures: Average volumes of operations between January and March 2018 by country.**

|  | All | Tanzania | | Malawi | | Zambia | |
|---|---|---|---|---|---|---|---|
|  | Mean | Mean | Std. Dev. | Mean | Std. Dev. | Mean | Std. Dev. |
| **Volume_6PC** | 140.5 | 124.2 | 49.3 | 225.5 | 107.5 | 65.6 | 47.9 |
| **Volume%** | 89.7% | 89.8% | | 88.9% | | 92.7% | |
| **Components** | Mean | Mean | V% | Mean | V% | Mean | V% |
| **caesarean sections** | 114.9 | 105.7 | 85.1% | 187.7 | 83.2% | 46.7 | 71.2% |
| **Hernia** | 8.3 | 6.0 | 4.8% | 12.3 | 5.5% | 6.2 | 9.4% |
| **Hydrocele** | 2.0 | 2.5 | 2.0% | 2.8 | 1.2% | 0.7 | 1.1% |
| **Hysterectomy** | 3.0 | 2.7 | 2.2% | 5.0 | 2.2% | 1.3 | 2.0% |
| **Laparotomy** | 5.1 | 4.3 | 3.5% | 7.2 | 3.2% | 3.7 | 5.6% |
| **Fracture** | 7.1 | 2.9 | 2.4% | 10.5 | 4.6% | 7.0 | 10.7% |

Note: Volume% refers to the share of six most common procedures (Volume_6PC) to the total number of all major (done on the operating theatre under anaesthesia) surgical procedures. V% refers to the share of each surgical procedure to the number of six procedures.

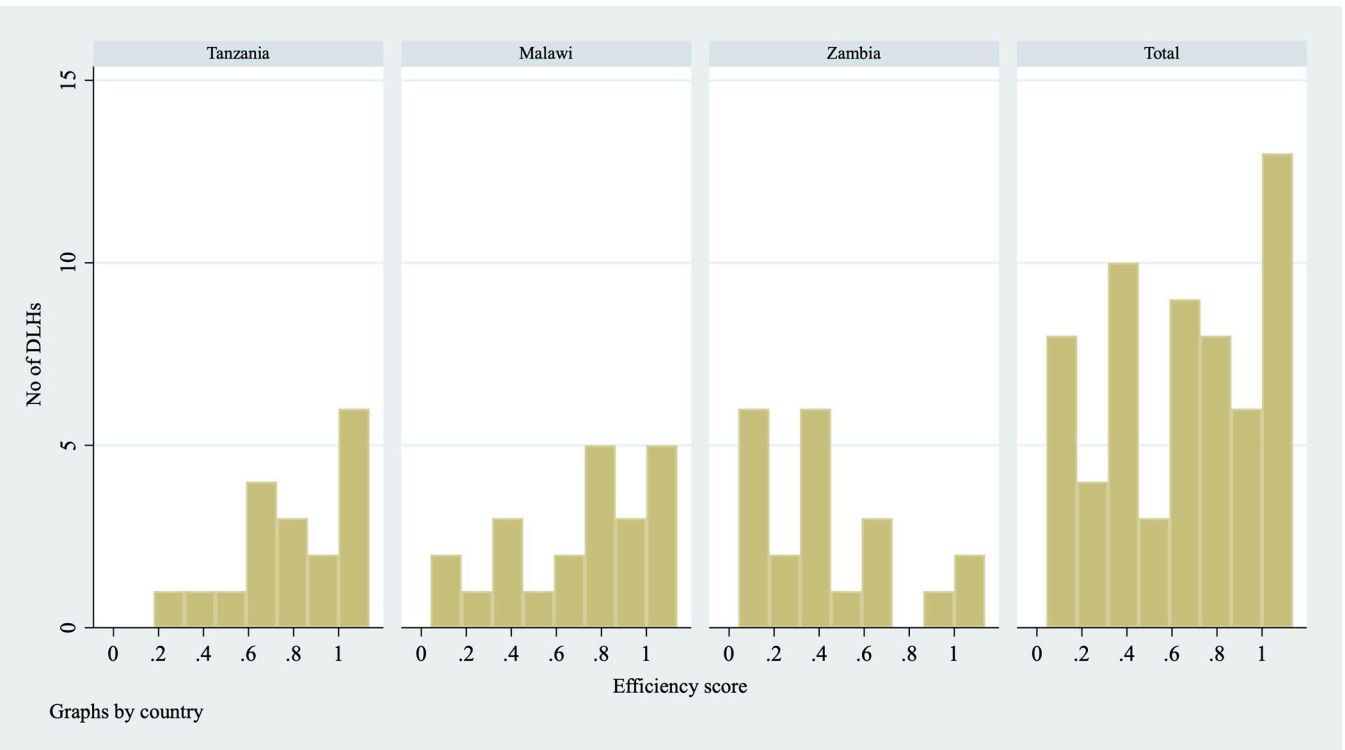

**Fig 1. Histogram of efficiency score by country.**

Fractures, at 10.7% (148/1377) of all procedures in Zambia, account for twice the proportion of procedures compared to Malawi and four times compared to Tanzania.

**Hospital efficiency.** Details of the DEA results by hospital and by country are provided in S1 and S2 Tables. The average efficiency score is the highest in Tanzania (0.774), followed by Malawi (0.705); it is considerably lower in Zambia (0.406). Fig 1 shows the frequency distributions of hospital efficiency scores by country. Of the eighteen DLHs in Tanzania, seven (39%) show high efficiency (DEA score = 1), compared with 9.5% (2/21) of DLHs in Zambia, where 71% (15/21) of DLHs score low on efficiency (DEA score<0.5). There are fewer low efficiency hospitals in Malawi and Tanzania (six and three, respectively). The lowest efficiency score is reported for a DLH in Zambia.

Table 4 shows the average value of production factors by country. DLHs with high efficiency scores have significantly more surgical staff compared with low efficiency hospitals

**Table 4. Average value of production factors by efficiency and country.**

| Country | Efficiency | Personnel | Infrastructure | Procedures | Equipment | Supplies | N |
|---|---|---|---|---|---|---|---|
| **Tanzania** | High | 11.3 | 8.8 | 24.8 | 17.3 | 15.7 | 6 |
| | Low | 9.8 | 9.2 | 25.5 | 15.6 | 17.2 | 12 |
| **Malawi** | High | 22.4 | 7.4 | 29.2 | 16.2 | 13.4 | 5 |
| | Low | 17.6 | 7.5 | 28.6 | 17.5 | 16.0 | 17 |
| **Zambia** | High | 6.5 | 9.5 | 22.0 | 18.0 | 10.5 | 2 |
| | Low | 4.5 | 9.3 | 26.1 | 18.9 | 19.3 | 19 |

Note: Hospitals are divided into two categories based on the DEA efficiency score: high efficiency (e = 1) and low efficiency (e<1). N is the number of DLHs in each category.

**Table 5. Relative weights of production factors using CSW-DEA.**

| Full sample | | Personnel | Infrastructure | Procedures | Equipment | Supplies |
|---|---|---|---|---|---|---|
| PIPES | | 1 | 0.8 | 2.4 | 1.3 | 1.5 |
| DEA | | 1 | 2.4 | 1.0 | 1.5 | 1.4 |
| By country | Volume | Personnel | Infrastructure | Procedures | Equipment | Supplies |
| Tanzania | 1 | 2.1 | 5.1 | 1.8 | 2.7 | 2.4 |
| Malawi | 1 | 3.3 | 9.0 | 3.3 | 5.1 | 5.1 |
| Zambia | 1 | 7.0 | 4.5 | 1.8 | 2.8 | 2.6 |

Note: The relative weights of input parameters using the full sample was divided by the weight of the personnel. The relative weight of inputs by country was divided by the weight of the output, the surgical volume.

(DEA score<1). The value of infrastructure does not differentiate between DLHs in terms of their efficiency scores. The average values of procedures and equipment for high efficiency hospitals are around the same or slightly lower than the values for low efficiency hospitals. Supplies are considerably lower in high efficiency compared to low efficiency hospitals in Zambia.

The comparison between the weights of production factors in the PIPES tool and the optimal set of input weights generated using the CSW-DEA is shown in Table 5. Using the CSW-DEA, the infrastructure has the highest weight, while according to the PIPES method of computing weights, the procedures have the highest weight. For the full sample, equipment and supplies are the other two important production factors, with similar weights in the two methods. In the cross-country analysis, infrastructure is the production factor with the highest weight, except in Zambia where the personnel has the highest weight for both methods used. In each country, the weight of the procedures is the lowest.

**Comparison of surgical capacity, productivity and efficiency.** Fig 2 presents the distribution of three core indicators (PIPES score, volume of procedures, and efficiency score) for the study hospitals in each country. The distribution of surgical capacity (PIPES index score) contrasts with the distribution of productivity (surgical volumes) and hospital efficiency (DEA score). Regions with comparatively lower capacity scores tend to have higher productivity and efficiency. For instance, the Western province in Zambia reports the highest PIPES score (7.95), the lowest surgical volume (26) and the lowest efficiency score (0.14). Specifically, Malawi reports the highest surgical productivity, while Zambia reports the lowest. The distribution of productivity is close to that of hospital efficiency.

Fig 3 compares the rankings of DLHs in the three countries for surgical capacity (PIPES), surgical productivity (volume of operations) and efficiency (DEA score). For most DLHs, the ranking of capacity is contrary to that of productivity. Across all three countries, hospitals with low rankings for surgical volume have similarly low rankings for efficiency and higher rankings for surgical capacity. In Malawi, there are eight hospitals with the same ranking for productivity and efficiency, seven of which rank lowest for both measures. Two DLHs in Tanzania (hospital 136 and 130) and three in Zambia (hospital 301, 318, and 306) rank the lowest in terms of both productivity and efficiency, with those in Zambia ranking remarkably high for capacity.

**Surgical productivity and production factors.** The relationship between surgical productivity (volume of operations) and production factors using the quantile regression is presented in S3 Table. Overall, personnel is positively and significantly correlated with surgical volume. District hospitals with more surgically active staff are more likely to be more productive. No significant correlation between surgical volume and other production factors is found. After

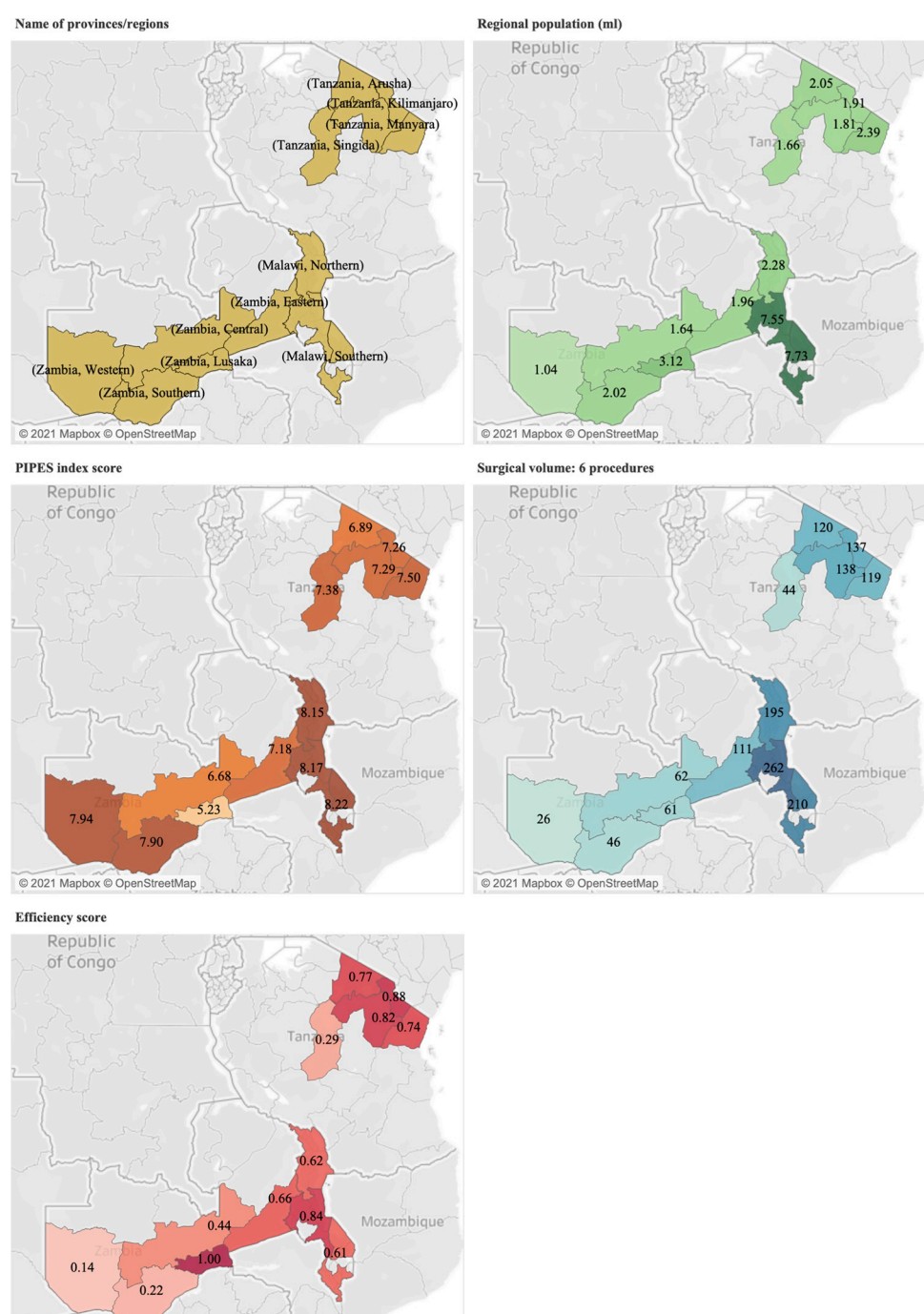

**Fig 2. Comparison of core indicators by country and region.** Yellow, names of provinces/regions. Green, regional population (million). Brown, PIPES index scores. Teal, surgical volume. Red, efficiency score. Information of regional population adapted from 2018 Population and Housing Census [32] (p12)], 2019 Tanzania in Figures [33] (p19)], and Population and Demographic Projections 2011–2035 [34] (p31, 32)]. Reprinted from Tableau Desktop 2020 under a CC BY license, with permission from © 2021 Mapbox © OpenStreetMap.

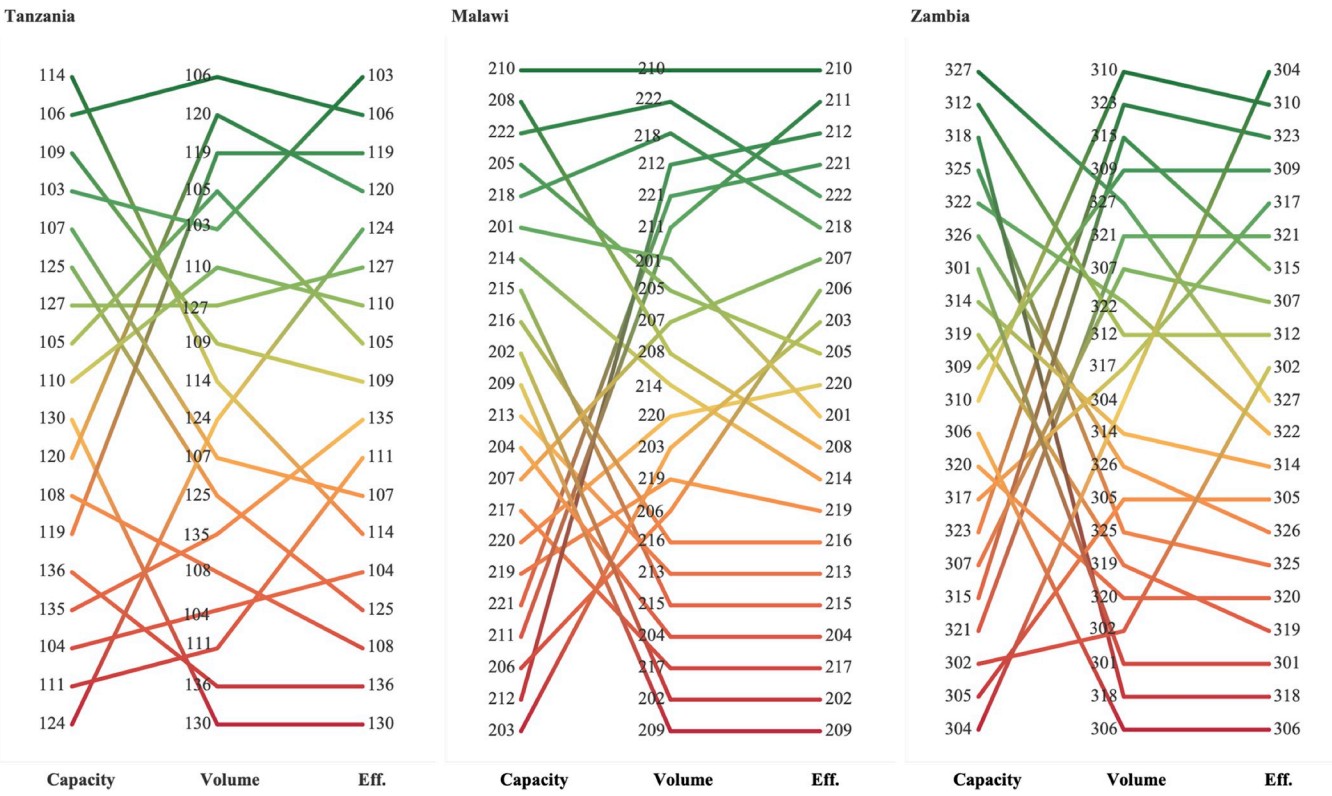

**Fig 3. Ranking of surgical capacity, productivity, and efficiency by hospital and country.** Green, high ranking of PIPES capacity score, surgical volumes, and hospital efficiency scores. Yellow, middle ranking of PIPES capacity score, surgical volumes, and hospital efficiency scores. Red, low ranking of PIPES capacity score, surgical volumes, and hospital efficiency scores. Rank of the PIPES capacity index, the surgical volume of six procedures, and hospital efficiency from the highest (green) to the lowest (red). The PIPES index score includes NPCs.

controlling for country specific characteristics, Malawi has higher surgical volumes than Tanzania, while Zambia reports significantly lower surgical volumes than Tanzania at the 75th percentile.

**Hospital efficiency and production factors.** Estimates of the relationship between the efficiency scores and production factors are shown in S4 Table. Results of the Tobit regression model use externally generated efficiency scores. The personnel is positively and significantly correlated with efficiency, a result that is maintained after controlling for country specific characteristics.

Results of the two-stage bootstrap model use bias-corrected efficiency scores. The personnel remained to be positively and significantly correlated with efficiency without controlling for country specific characteristics. At the country level, the numbers of procedures and equipment are weakly and negatively correlated with efficiency, while the number of supplies is weakly and positively correlated with efficiency. Zambia shows much lower efficiency scores than Tanzania.

## Discussion

Our findings demonstrate that current surgical capacity assessment measures are not reliable predictors of district level hospital surgical productivity or efficiency and—if used alone—are not suitable for planning surgical scale-up, or surgical services scaling in general. Several studies have measured surgical capacity alone (using PIPES or similar instruments) and made

recommendations for investments in resources/inputs, implicitly assuming that more inputs lead almost automatically to greater surgical productivity [17, 35, 36]. Our findings question this assumption, demonstrating that surgical production factors do not contribute equally to outputs; and there can even be an inverse relationship between surgical capacity and efficiency. All hospitals that scored high on surgical capacity (PIPES index) scored low for efficiency, which suggests that the provision of more resources to a hospital may not make it more productive. Instead, there may be a waste of resources unless, as in the case of Malawi, such additional resources consist of surgically active personnel. An earlier study in Tanzania [37] also demonstrated that clinicians' surgical skills and staff skills-mix are major factors that determine efficiency of resource use and economies of scale. However improvements in surgical skills will not result in increased volume and efficiency if other rate-limiting steps are not removed simultaneously.

Tanzania had the most efficient hospitals in this three country sample. However, there are no standardised criteria that define the surgical efficiency of DLHs in SSA. The reported high capacity in Malawi was mainly driven by the relatively high number of surgically active NPCs, but in respect to all other capacity inputs Malawi scored lowest. Despite infrastructure, supplies and equipment shortages, Malawi [38] had the highest surgical productivity, twice that of Tanzania and over four times higher than in Zambia. This may reflect the higher population demand for major surgery at district level in Malawi [39] and the relatively small number of DLHs serving its population. The Malawi findings are consistent with the core finding that investment in the surgical workforce is more critical than infrastructural inputs or supplies. At the same time, however, it is important to realise that investing in expansion of the surgical workforce only may produce just a short-term increase in productivity. If workload, staff motivation, and sustainable financing of surgery are not also addressed, DLHs may not be able to sustain high surgical outputs [40].

Hospitals in Zambia had the lowest efficiency, but comparatively high PIPES capacity scores, suggesting that despite good availability of infrastructure, equipment and supplies, operating theatres were underutilised. This corresponds with our earlier study, which reported particularly poor utilisation of operating theatres in Zambia, at around 10% of capacity [35]. Zambia recently launched a national programme to build an additional 115 'mini hospitals', with bed capacities of 80 and an operating theatre to perform basic surgery [41]. Our findings suggest that this may not be the most efficient approach to meeting the needs of dispersed rural populations. Given that surgical productivity and hospital efficiency in Zambia are lower than in the other two countries, it is doubtful if the existing size of the surgical workforce [42] can justify this expansion, or if there will be demand to meet the increased supply. The shortage of surgical and anaesthesia staff is already affecting existing facilities in Zambia [43]. Given the low density in rural Zambia, there may be a better case for investing in ambulance services to transfer patients to higher-level hospitals with sufficient surgical staff capacity and case volume for absorbing more cases.

Weights of production factors, used to identify the factor with greater influence on hospital efficiency, show a different emphasis in practice. The weight of the procedures (self-declared ability to perform certain surgical procedures) used in the PIPES was highest; while the infrastructure weight was highest when DEA was used. If national prioritisation and planning exercises and investments rely on simple capacity assessments, they will overestimate the importance of the procedures production factor as a measure and means for achieving an increase in surgical capacity. Although self-reported capacity suggests that the surgical providers in our study were trained to conduct more complex operations than the core six surgical procedures examined in this study, the core skill set most commonly utilised by them was obstetric surgery-related, especially in Tanzania and Zambia. Hence, there is a case for using

the volume and spectrum of actual procedures undertaken to define the training package for district level surgery in SSA countries to avoid underutilisation of capacity that does not match the demand. This needs to be accompanied by appropriate referral guidelines to ensure continuity of care.

In all countries, averages for procedures, equipment, and supplies were lower for hospitals with the highest efficiency score, meaning that efficient hospitals had fewer supplies available to conduct more operations compared to inefficient hospitals. In contrast, the findings suggest there may be an excess of supplies in low-efficiency hospitals, which in the presence of surgical staff shortages may lead to waste or, alternatively, contribute to excess of surgical workloads and burnout on scarce staff. These unintended effects from strengthening equipment and supplies require further exploration. For now, policy makers and hospital managers should re-evaluate the rationale for and the allocation of surgical resources across regions and facilities, in relation to surgical productivity and demand; and consider ways to improve surgical system efficiencies.

There are some limitations to this paper. First is the absence of data on surgical quality in the hospital efficiency analysis. Data on surgical quality were not available in this study. The mortality rate in the sample was very low (0.08%), hence it is excluded in the DEA. Secondly, the analysis does not include expenditure on surgery that may have an effect on all measured aspects. Although DLHs have a series of income sources varied by poverty, to our knowledge district hospitals have similar financing structures within countries, so expenditure on surgery should not differentiate hospitals significantly. Thirdly, this study was based on the observation conducted for a single period of time. Surgical volume, capacity and efficiency can change over time, and in future studies we recommend including multiple time points of data collection (longitudinal study design) to capture those potential changes. A further limitation is the relatively small sample size of hospitals from which data were collected. The estimation of hospital efficiency would have been more accurate if the sample size was larger, hence the use of the bootstrap method to minimise its impact.

## Conclusion

Studies such as this one, where three aspects of surgical performance (capacity, productivity and efficiency), are investigated at the same time provide the needed evidence about where inefficiencies are in the surgical system. However more research is needed to identify an optimal method of assessing capacity, productivity and efficiency in surgical care in order to improve planning. In all countries our analysis highlights the importance of adequate human resources in the delivery of surgical care in district hospitals, which presents strong and positive correlations with surgical productivity and hospital efficiency. Surgical training at non-specialist level [4–7] can be prioritised in district hospitals in order to improve hospital efficiency and responsiveness to the growing demand for surgical care. This finding can be important for countries which developed NSOAPs with limited evidence from empirical data.

## Supporting information

**S1 Table. Full result of DEA by country.**
(PDF)

**S2 Table. Average value of input slacks by efficiency and country.**
(PDF)

**S3 Table. Relationship between surgical productivity and production factors–quantile regression at 25th, 50th, and 75th percentiles.**
(PDF)

**S4 Table. Relationship between surgical efficiency and production factors using Tobit and two-stage bootstrap models.**
(PDF)

**S1 File. Data envelopment analysis.**
(PDF)

**S2 File. Data for analysis.**
(ZIP)

## Author Contributions

**Conceptualization:** Mengyang Zhang, Jakub Gajewski, Chiara Pittalis, Mark Shrime.

**Data curation:** Mengyang Zhang, Jakub Gajewski, Chiara Pittalis, Morgane Clarke, Adinan Juma, Gerald Mwapasa.

**Formal analysis:** Mengyang Zhang, Mark Shrime.

**Funding acquisition:** Jakub Gajewski, Eric Borgstein, Chris Lavy, Nyengo Mkandawire, John Kachimba, Michael Mbambiko, Kondo Chilonga, Leon Bijlmakers, Ruairi Brugha.

**Investigation:** Mengyang Zhang, Jakub Gajewski, Chiara Pittalis, Mark Shrime, Henk Broekhuizen, Martilord Ifeanyichi, Eric Borgstein, Nyengo Mkandawire, John Kachimba, Michael Mbambiko, Kondo Chilonga, Leon Bijlmakers, Ruairi Brugha.

**Methodology:** Mengyang Zhang, Mark Shrime.

**Project administration:** Jakub Gajewski, Chiara Pittalis, Henk Broekhuizen, Morgane Clarke, Eric Borgstein, Grace Drury, Adinan Juma, Nyengo Mkandawire, Gerald Mwapasa, John Kachimba, Michael Mbambiko, Kondo Chilonga, Leon Bijlmakers, Ruairi Brugha.

**Resources:** Jakub Gajewski, Chiara Pittalis.

**Software:** Mengyang Zhang, Mark Shrime.

**Supervision:** Jakub Gajewski, Eric Borgstein, Chris Lavy, Nyengo Mkandawire, John Kachimba, Michael Mbambiko, Kondo Chilonga, Leon Bijlmakers, Ruairi Brugha.

**Validation:** Mengyang Zhang, Jakub Gajewski, Chiara Pittalis, Henk Broekhuizen, Martilord Ifeanyichi, Leon Bijlmakers, Ruairi Brugha.

**Visualization:** Mengyang Zhang.

**Writing – original draft:** Mengyang Zhang, Jakub Gajewski, Chiara Pittalis.

**Writing – review & editing:** Mengyang Zhang, Jakub Gajewski, Chiara Pittalis, Mark Shrime, Henk Broekhuizen, Martilord Ifeanyichi, Morgane Clarke, Eric Borgstein, Chris Lavy, Grace Drury, Adinan Juma, Nyengo Mkandawire, Gerald Mwapasa, John Kachimba, Michael Mbambiko, Kondo Chilonga, Leon Bijlmakers, Ruairi Brugha.

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
