## [Decision Letter · Decision Letter 0]

30 Jun 2022

PONE-D-22-07365Surgical capacity, productivity and efficiency at the district level in Sub-Saharan Africa: a three-country studyPLOS ONE

Dear Dr. Zhang,

Thank you for submitting your manuscript to PLOS ONE. After careful consideration, we feel that it has merit but does not fully meet PLOS ONE’s publication criteria as it currently stands. Therefore, we invite you to submit a revised version of the manuscript that addresses the points raised during the review process.

We look forward to receiving your revised manuscript.

Kind regards,

Majid Soleimani-damaneh

Academic Editor

PLOS ONE

Journal Requirements:

2. “Please include a complete copy of PLOS’ questionnaire on inclusivity in global research in your revised manuscript. Our policy for research in this area aims to improve transparency in the reporting of research performed outside of researchers’ own country or community. The policy applies to researchers who have travelled to a different country to conduct research, research with Indigenous populations or their lands, and research on cultural artefacts. The questionnaire can also be requested at the journal’s discretion for any other submissions, even if these conditions are not met.  Please find more information on the policy and a link to download a blank copy of the questionnaire here: https://journals.plos.org/plosone/s/best-practices-in-research-reporting. Please upload a completed version of your questionnaire as Supporting Information when you resubmit your manuscript

3. In ethics statement in the manuscript and in the online submission form, please provide additional information about the patient records/samples used in your retrospective study. Specifically, please ensure that you have discussed whether all data/samples were fully anonymized before you accessed them and/or whether the IRB or ethics committee waived the requirement for informed consent. If patients provided informed written consent to have data/samples from their medical records used in research, please include this information.

“The project is funded by the European Union’s Horizon 2020 research and innovation programme under the grant agreement No. 733391. Salaries of JG, MZ, and CP were paid by this grant. Cost of the study was covered by this grant.”

6. Please include captions for your Supporting Information files at the end of your manuscript, and update any in-text citations to match accordingly. Please see our Supporting Information guidelines for more information: http://journals.plos.org/plosone/s/supporting-information

7. We note that Figure 2 in your submission contain map images which may be copyrighted. All PLOS content is published under the Creative Commons Attribution License (CC BY 4.0), which means that the manuscript, images, and Supporting Information files will be freely available online, and any third party is permitted to access, download, copy, distribute, and use these materials in any way, even commercially, with proper attribution. For these reasons, we cannot publish previously copyrighted maps or satellite images created using proprietary data, such as Google software (Google Maps, Street View, and Earth). For more information, see our copyright guidelines: http://journals.plos.org/plosone/s/licenses-and-copyright.

   a. You may seek permission from the original copyright holder of to Figure 2 publish the content specifically under the CC BY 4.0 license. 

Reviewers' comments:

Reviewer's Responses to Questions

**Comments to the Author**

1. Is the manuscript technically sound, and do the data support the conclusions?

Reviewer #1: Yes

Reviewer #2: Partly

2. Has the statistical analysis been performed appropriately and rigorously? 

Reviewer #1: Yes

Reviewer #2: I Don't Know

3. Have the authors made all data underlying the findings in their manuscript fully available?

Reviewer #1: Yes

Reviewer #2: No

4. Is the manuscript presented in an intelligible fashion and written in standard English?

Reviewer #1: Yes

Reviewer #2: Yes

5. Review Comments to the Author

Reviewer #1: This paper deals with the determinants of hospital efficiency in district hospitals in three African countries, Tanzania, Malawi, and Zambia.

The authors determine the hospital efficiency scores for three counties. They discussed that hospital efficiency, combining capacity measures and surgical outputs, is a better indicator of surgical performance than capacity measures, which could be misleading if used alone for surgical planning. They concluded that investment in the surgical workforce, in particular, plays a central role in improving district hospital surgical productivity and efficiency.

Overall, the aim of this paper may be attractive for the practitioner of the DEA model; but first, the authors should address the following comments.

One of the features of the DEA models is that of providing the benchmark (target unit) for inefficient units, While the authors do not discuss the key concept of target unit for inefficient DMUs. The authors must suggest the target unit for inefficient DMUs to learn and improve by following it.

P. 6 L. 1 Hospital efficiency

The authors obtain the efficiency scores using the constant returns to scale (CRS) DEA model while utilizing the variable returns to scale (VRS) DEA model would be more common.

P. 6 L. 7 Typo

Chames et al. → Charnes et al.

P. 12 L. 2 Congestion in the resources

The authors state as follows:

The above statements show that the congestion may be present in the inputs. The authors must deal with identifying and managing congestion using DEA.

P. 14 L. 6 The mortality rate

The authors state as follows:

“The mortality rate in the sample was very low (0.08%), hence it is excluded in the DEA.” The mortality rate in the concept of DEA is an undesirable output that should be considered in the efficiency score of the hospitals.

S1 Table. Full result of DEA by country

The authors only identify the returns to scale (RTS) status of hospitals without any interpretation. Why do all DMUs show constant RTS or increasing RTS? The authors must discuss this concept in detail.

Reviewer #2: This paper presents the application of data envelopment analysis (DEA) in the hospital ward. In fact, this paper investigates the determinants of hospital efficiency in district hospitals in three African countries. For this purpose, the initial DEA models (CCR, BCC) have been used.

Here are some of my comments on the paper:

1) The DEA literature needs to be better articulated. References need to be updated.

2) Explain the reasons for choosing production indicators. Input factors must also be stated.

3) For better results, I suggest using the SBM-model as well

6. PLOS authors have the option to publish the peer review history of their article (what does this mean?). If published, this will include your full peer review and any attached files.

Reviewer #1: **Yes: **Amin Mostafaee

Reviewer #2: No

---

## [Author Response · Author response to Decision Letter 0]

11 Aug 2022

Reviewer 1

1. This paper deals with the determinants of hospital efficiency in district hospitals in three African countries, Tanzania, Malawi, and Zambia.

The authors determine the hospital efficiency scores for three counties. They discussed that hospital efficiency, combining capacity measures and surgical outputs, is a better indicator of surgical performance than capacity measures, which could be misleading if used alone for surgical planning. They concluded that investment in the surgical workforce, in particular, plays a central role in improving district hospital surgical productivity and efficiency.

Overall, the aim of this paper may be attractive for the practitioner of the DEA model; but first, the authors should address the following comments:

One of the features of the DEA models is that of providing the benchmark (target unit) for inefficient units, While the authors do not discuss the key concept of target unit for inefficient DMUs. The authors must suggest the target unit for inefficient DMUs to learn and improve by following it.

Thank you for the comments. There is no reference point for inefficient facilities, hence our approach needed to be adjusted. Rather than focusing on specific inefficient hospitals, we highlight the problems of hospitals with low efficiency (DEA score < 0.5). As what’s shown in Table 4, compared with highly efficient hospitals, those hospitals had lower values scores in surgical staff but relatively higher score in supplies, which were consistent in three countries. Hence, in Discussion we suggested that investment in surgical staff would be a better choice for the national surgical planning.

2. P. 6 L. 1 Hospital efficiency

The authors obtain the efficiency scores using the constant returns to scale (CRS) DEA model while utilizing the variable returns to scale (VRS) DEA model would be more common.

Thank you for your comments. Yes, the use of VRS can provide additional details. Due to the text length requirement, we haven’t included the VRS results in the main context but in the Supporting Information where we also presented the details of efficiency score using variable returns to scale and the nonincreasing returns to scale.

3. P. 6 L. 7 Typo

Chames et al. → Charnes et al.

Thank you for pointing it out. It has been corrected. 

4. P. 12 L. 2 Congestion in the resources

The authors state as follows:

“Our findings question this assumption, demonstrating that surgical production factors do not contribute equally to outputs; and there can even be an inverse relationship between surgical capacity and efficiency. All hospitals that scored high on surgical capacity (PIPES index) scored low for efficiency, which suggests that the provision of more resources to a hospital may not make it more productive.”

The above statements show that the congestion may be present in the inputs. The authors must deal with identifying and managing congestion using DEA.

Thank you for your comments. The DEA efficiency suggests the best allocation of different inputs under certain assumptions given the current output level. Slacks that indicate the over input will lead to inefficiency. Due to the length requirement, we haven’t included the slacks discussion in the paper, because the core interest of this paper is to show the difference in measuring hospital performance (the difference in input weights using different measurement), thereby drawing more attention to the measurement selection and healthcare development at the district level. In addition, the common set of weights (CSW) in DEA was included to improve the accuracy of the input weights, the result of which was shown in Table 5. The result of input slacks are added into S3 Data envelopment analysis result in Supporting Information. 

5. P. 14 L. 6 The mortality rate

The authors state as follows:

“The mortality rate in the sample was very low (0.08%), hence it is excluded in the DEA.” The mortality rate in the concept of DEA is an undesirable output that should be considered in the efficiency score of the hospitals.

We would like to point out that the low mortality rate is very low in the sample of hospitals in our study, and it is rather homogeneous across the sample. If 99.2% of DMUs have no morality rate, then there is little difference in the output (mortality) and we cannot run the DEA regression for a reliable result. 

6. S1 Table. Full result of DEA by country

The authors only identify the returns to scale (RTS) status of hospitals without any interpretation. Why do all DMUs show constant RTS or increasing RTS? The authors must discuss this concept in detail.

Thank you for your comments. We have added an explanation for the DEA results. Table S1 shows the efficiency score under the constant returns to scale, variable returns to scale, and the nonincreasing returns to scale. The Stata command also calculates the returns to scale (the last column of Table S1) by calculating the marginal productivity. If the marginal productivity equals to 1, it is IRS. If the marginal productivity equals to 0, it is CRS. All related outputs were shown in Table S1. The explanation about the marginal values has been added to the table note. 

 

Reviewer 2

This paper presents the application of data envelopment analysis (DEA) in the hospital ward. In fact, this paper investigates the determinants of hospital efficiency in district hospitals in three African countries. For this purpose, the initial DEA models (CCR, BCC) have been used.

Here are some of my comments on the paper:

1) The DEA literature needs to be better articulated. References need to be updated.

2) Explain the reasons for choosing production indicators. Input factors must also be stated.

3) For better results, I suggest using the SBM-model as well

Thank you for your comments. We have addressed them in the new version of the manuscript:

1) Except the reference of the methodology and command, other DEA related references are replaced by the new publications.

2) The production factors are the factors used in the PIPES index survey defined by an organisation name Surgeons Overseas. In order to compare the difference in using different measures for measuring production performance, these five production factors were included in the DEA regression. The output factor used in the DEA is the surgical volume, which is also widely used as an indicator of hospital performance. As we explained in the paper, the mortality rate is too low to include in the DEA. The surgical volume is the only output factor for which we have collected data. 

3) The fitness and efficiency of the DEA model was considered when we prepared the manuscript. Due to the manuscript length requirement, it was not possible to provide all details. The comment set of weights (CSW) method was used to calculate the optimal input weights by setting the bounds of variables and improving nonlinearity. More details were shown in the Supporting Information.

---

## [Decision Letter · Decision Letter 1]

5 Oct 2022

PONE-D-22-07365R1Surgical capacity, productivity and efficiency at the district level in Sub-Saharan Africa: a three-country studyPLOS ONE

Dear Dr. Zhang,

Thank you for submitting your manuscript to PLOS ONE. After careful consideration, we feel that it has merit but does not fully meet PLOS ONE’s publication criteria as it currently stands. Therefore, we invite you to submit a revised version of the manuscript that addresses the points raised during the review process.

We look forward to receiving your revised manuscript.

Kind regards,

Majid Soleimani-damaneh

Academic Editor

PLOS ONE

Reviewers' comments:

Reviewer's Responses to Questions

**Comments to the Author**

1. If the authors have adequately addressed your comments raised in a previous round of review and you feel that this manuscript is now acceptable for publication, you may indicate that here to bypass the “Comments to the Author” section, enter your conflict of interest statement in the “Confidential to Editor” section, and submit your "Accept" recommendation.

Reviewer #1: (No Response)

Reviewer #2: (No Response)

2. Is the manuscript technically sound, and do the data support the conclusions?

Reviewer #1: Partly

Reviewer #2: (No Response)

3. Has the statistical analysis been performed appropriately and rigorously? 

Reviewer #1: (No Response)

Reviewer #2: (No Response)

4. Have the authors made all data underlying the findings in their manuscript fully available?

Reviewer #1: Yes

Reviewer #2: (No Response)

5. Is the manuscript presented in an intelligible fashion and written in standard English?

Reviewer #1: Yes

Reviewer #2: (No Response)

6. Review Comments to the Author

Reviewer #1: The authors have incorporated some comments from the first round of review, but I am not convinced by the responses to the comments # 1 and # 4.

Reviewer #2: The authors have considered all my points in the revised article, so I think the revised article can be accepted.

7. PLOS authors have the option to publish the peer review history of their article (what does this mean?). If published, this will include your full peer review and any attached files.

Reviewer #1: **Yes: **Amin Mostafaee

Reviewer #2: No

---

## [Author Response · Author response to Decision Letter 1]

26 Oct 2022

Reviewer 1

The authors have incorporated some comments from the first round of review, but I am not convinced by the responses to the comments # 1 and # 4.

Thank you for your comments. Those two comments should be related to target units for inefficient DMUs and resources congestion. 

Responding to the first point, we would like to reiterate that this study aimed to compare the difference between hospital performance measures by examining the determinant of hospital efficiency using data envelopment analysis (DEA). To ensure consistency in the analyses, the production factors used in the PIPES capacity index were used in the DEA. 

We found that one reason the PIPES index and hospital efficiency presented almost opposite results was different weights assigned to the production factors. As what is shown in Table 5, the PIPES capacity index assigns higher weights to procedures, equipment and supplies that are less likely to be improved in a short time. DEA method weights infrastructure and personnel higher.

As to the ‘target units for inefficient DMUs’, we think what the reviewer meant was the efficient hospital with DEA score equal to one. Table 4 compared the difference in each production factor between high-efficiency hospitals (DEA score=1) and low-efficiency hospitals (DEA score<1), which provided information about ‘the target unit for inefficient DMUs to learn’. However, surgical efficiency is different from production efficiency. According to the production theory, any input, such as capital, wage, and raw materials, can be divided by units and then change the amount of inputs to produce the final good most efficiently. Surgical capacity, in turn, is hard to change. In Table 4, low-efficiency hospitals had higher infrastructure, procedures, equipment, and supplies than efficient ones. In theory, these inputs should be reduced in production for higher efficiency. But in practice, it is almost impossible to reduce infrastructure, equipment or supplies, and advanced skills surgeons have learned. Therefore, the only one possible way is to increase the personnel, i.e. more surgeons, doctors, and NPCs, since low-efficiency hospitals have lower personnel than efficient ones. And training NPCs has been proven as a good way in the literature to improve the number of surgically trained cadres in settings such as our study. That is why the conclusion was that “… investment in the surgical workforce, in particular, plays a central role in improving district hospital surgical productivity and efficiency”.

In response to the point about congestion, similar assumptions apply. Those district hospitals are the frontline providers of surgical services in rural areas. And Figure 3 illustrated that the performance results measured by efficiency score and surgical volume were similar. If more personnel can become the one input increased in those low-efficiency hospitals, then the slacks of other types of production factors will not be a restriction of efficiency but help to improve the output – providing more surgical services to rural populations.

 

Reviewer 2

The authors have considered all my points in the revised article, so I think the revised article can be accepted.

Thank you so much for your comments.

---

## [Decision Letter · Decision Letter 2]

14 Nov 2022

Surgical capacity, productivity and efficiency at the district level in Sub-Saharan Africa: a three-country study

PONE-D-22-07365R2

Dear Dr. Zhang,

We’re pleased to inform you that your manuscript has been judged scientifically suitable for publication and will be formally accepted for publication once it meets all outstanding technical requirements.

Kind regards,

Majid Soleimani-damaneh

Academic Editor

PLOS ONE

---

## [Editor Report · Acceptance letter]

18 Nov 2022

PONE-D-22-07365R2 

Surgical capacity, productivity and efficiency at the district level in Sub-Saharan Africa: a three-country study 

Dear Dr. Zhang:

I'm pleased to inform you that your manuscript has been deemed suitable for publication in PLOS ONE. Congratulations! Your manuscript is now with our production department. 

Kind regards, 

on behalf of

Dr. Majid Soleimani-damaneh 

Academic Editor

PLOS ONE